# Hydrogeological Parameter Estimation of Confined Aquifer within a Rectangular Shaped Drop Waterproof Curtain

**Yi Li** [1], **Wentao Xie** [1], **Hongwei Wang** [1], **Bin Peng** [1], **Feng Xiong** [2,*] and **Chun Zhu** [3]

1. Powerchina Hubei Electric Engineering Co., Ltd., Wuhan 430040, China
2. Faculty of Engineering, China University of Geosciences (Wuhan), Wuhan 430074, China
3. School of Earth Sciences and Engineering, Hohai University, Nanjing 210098, China
* Correspondence: fengxiong@cug.edu.cn; Tel.: +86-13163244259

**Abstract:** For the dewatering of deep excavation, the existing man-made waterproof curtain has a significant influence on flow response in confined aquifers; the effect of the waterproof curtain must be considered when using the field data for hydrogeological parameter estimation. In this study, a closed-form analytical solution for constant discharge pumping in a confined aquifer within a rectangular-shaped drop waterproof curtain is obtained by making use of the image method coupled with the superpose principle. A straight-lined method is presented to determine the value of the hydraulic parameters of the confined aquifer and the application of the obtained results is illustrated by the usefulness of a field pumping test in Wuhan, China. The results show that the predicted drawdowns developed by the estimated parameters are in good agreement with the measured drawdown in the field. The proposed solution and parameter estimation are reliable and can provide important help for the design of dewatering in deep foundation pit engineering.

**Keywords:** confined aquifer; drop waterproof curtain; parameter estimation; pumping test

## 1. Introduction

A variety of existing analytic models for pumping-induced flow in different aquifer systems are obtained on the basis of the assumption that the aquifer is horizontal of infinite extent [1–7]. The assumption is only available when the pumping or injection does not spread to the nearby finite boundaries which are often described by constant-head or no-flow boundaries. Otherwise, it may not be appropriate for using the assumption of the aquifer to an infinite extent, and some errors can also be found [8–11]. Therefore, much attention has been paid to the effect of outer finite boundary and the analytical expression for predicting the drawdown induced by a discharging well in aquifers with a nearby bounded impermeable or recharge boundary can be observed in well hydraulic literature. For example, ref. [12] presented analytical solutions for periodic well recharge in rectangular aquifers with third-kind boundary conditions by using Laplace and finite Fourier transforms. Ref. [13] obtained an analytical drawdown solution for constant-flux pumping in a finite two-zone confined aquifer with an outer constant head condition. Ref. [14] investigated pumping in a rectangular coastal aquifer that is bounded by two parallel imperious boundaries and two parallel constant-head boundaries. Ref. [15] applied Schwartz–Christoffel conformal mapping method and the complex variable techniques, and the steady-state analytical solutions were given for constant rate pumping in a rectangular aquifer considering four different combinations of impermeable and constant-head boundary conditions. Recently, ref. [16] derived a general analytical solution for pumping tests in radial finite two-zone confined aquifers with a Robin-type outer boundary. Ref. [17] developed semi-analytical solutions for constant-head pumping in a finite leaky confined aquifer with an imperious or constant-head outer boundary. These studies demonstrated that the effect of boundary conditions on flow was not to be neglected.

It is known that the hydrogeological parameter (e.g., the transmissivity and storage coefficient (T and S)) of porous media is one of the important parameters, and their accuracy has some significant influences on the predictions for groundwater flow in different aquifer systems in groundwater science and engineering, on the design of the dewatering system in underground engineering, especially in deep excavation engineering [2,18–27]. Laboratory tests, field tests, and empirical formulas are often employed to determine their values. The most reliable method is the field tests, and a larger number of studies have focused on the determination of the hydrogeological parameter based on the collected in situ data such as drawdown and wellbore discharge during pumping tests [11,28]. For example, ref. [3] proposed the ratio method to determine the parameters of the aquifer and its adjacent aquitards in a multiple aquifer system. Ref. [29] estimated the hydraulic parameters for leaky aquifers using the extended Kalman filter. Ref. [4] performed analytical and numerical analyses using different models to estimate both saturated and unsaturated zone hydraulic parameters in unconfined aquifers. Ref. [30] performed a comparison for the estimation of leaky aquifer parameters obtained by using three different methods. Ref. [31] identified the hydraulic parameters of the confined aquifer by using the PEST and Theis model on the basis of the field variable pumping/injection tests. Ref. [32] gave a numerical method to identify hydrogeological parameters by minimizing the difference between measured and calculated values. Ref. [33] presented two graphical methods for the hydraulic parameter estimation of the tested confined aquifer with a pumping well having an exponentially decreasing rate [34] discussed the applicability of various models for the determination of different hydraulic parameters in a multiaquifer system and presented a new parameter estimation method named GALMA. Ref. [35] estimated the hydraulic parameters for an alluvial aquifer incorporating the geophysical survey, pumping tests, and simulation of hydrofacies model to provide a complete understanding of the aquifer characteristics. One can see that the above-mentioned studies are established on the assumption that the aquifer is of infinite extent in the horizontal direction, and the effect of the finite boundary is neglected.

In this study, we derive closed-form analytical solutions for pumping in a confined aquifer that is fully bounded by a rectangular-shaped waterproof curtain with the aid of the image method and superposition principle. A straight-lined method is then proposed to identify the hydrogeological parameters of the confined aquifer and the application of the obtained results is illustrated by its usefulness in a field pumping test. The derived results could offer guidance for designing dewater schemes in deep excavation considering the effect of fully penetrated waterproof curtains (boundaries).

## 2. Methods

### 2.1. Drawdown Solution for Flow in a Confined Aquifer of Infinite Extent

2.1.1. Mathematical Model

Figure 1 shows a constant-rate pumping well partially penetrated in a homogeneous confined aquifer of infinite extent. The well with a screening interval from d to d + l has an infinitesimal radius. The main aquifer is of uniform hydraulic conductivity (K) and thickness (M). Thus, one can obtain the mathematical model associated with the following boundary-value problem:

$$\frac{\partial^2 s(r,z,t)}{\partial r^2} + \frac{1}{r}\frac{\partial s(r,z,t)}{\partial r} + \frac{\partial^2 s(r,z,t)}{\partial^2 z} = \frac{S_s}{K}\frac{\partial s(r,z,t)}{\partial t} \tag{1}$$

$$s(r,z,0) = 0 \tag{2}$$

$$s(\infty,z,t) = 0 \tag{3}$$

$$\frac{\partial s}{\partial z}(r,0,t) = 0 \tag{4}$$

$$\frac{\partial s}{\partial z}(r,M,t) = 0 \tag{5}$$

$$\lim_{r \to 0} r \frac{\partial s}{\partial r} = \begin{cases} -\frac{Q}{2\pi K l}, & d \le z \le l + d \\ 0, & 0 < z < d \quad l + d < z < M \end{cases} \tag{6}$$

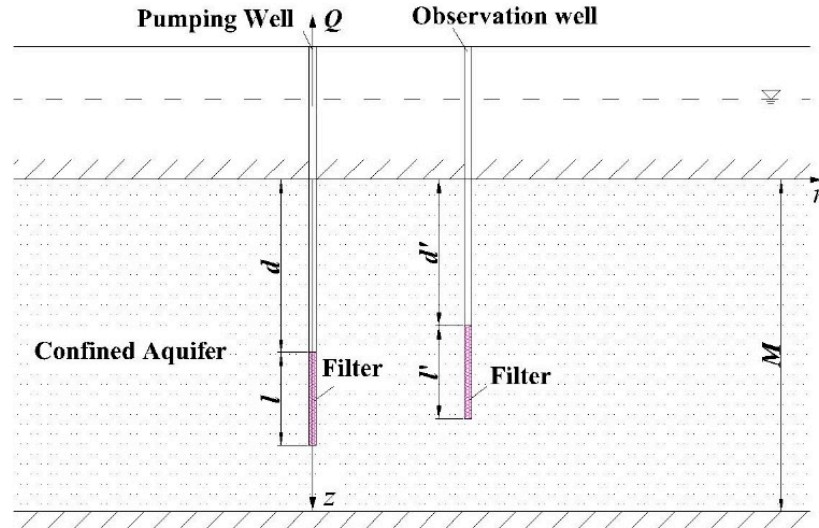

**Figure 1.** Schematic representation of an infinite confined aquifer partially penetrated by pumping and observation wells.

### 2.1.2. Solution

Hantush (1964) [2] solved the Equations (1)–(6) and obtained the following close-formed solution for drawdown in a piezometer:

$$s = \frac{Q}{4\pi T} \left\{ W(u) + \frac{2M}{\pi l} \sum_{n=1}^{\infty} \frac{1}{n} \left\{ \sin\left[ \frac{(l+d)n\pi}{M} \right] - \sin\left( \frac{n\pi d}{M} \right) \right\} \cdot \cos\left( \frac{n\pi z}{M} \right) \cdot W(u, x_n) \right\} \tag{7}$$

where $Q$ is pumping discharge, $T = KM$ is aquifer transmissivity; $l$ is the well screen length of the pumping well; $d$ is the vertical distance from the bottom of the overlain imperious layer to the top of the well screen of the pumping well; $W(u)$ refers to Theis well function and defined by

$$W(u) = \int_u^{\infty} \frac{1}{y} \exp(-y) dy = -0.5772 - \ln u + \sum_{n=1}^{\infty} (-1)^n \frac{u^n}{n \cdot n!} \tag{8}$$

and $W(u, x_n)$ is defined by

$$W(u, x_n) = \int_u^{\infty} \frac{1}{y} \exp\left( -y - \frac{x_n^2}{4y} \right) dy \tag{9}$$

in which $u = r^2 S/(4Tt)$, $x_n = n\pi r/M$, $Ss$ represents the specific storage of aquifer.

Notably, the available studies show that if the pumping time is long or $r$ is too small, the value of $u$ is less than 0.01 and then the Theis well function can be approximately written as [2,36–38]:

$$W(u) \approx -0.5772 - \ln u \tag{10}$$

and $W(u, x_n)$ can be approximately expressed by

$$W\left( u, \frac{n\pi r}{M} \right) \cong 2K_0\left( \frac{n\pi r}{M} \right) \tag{11}$$

in which $K_0(\cdot)$ is the zero-order modified Bessel function of the second kind.

Substituting $u = r^2S/4Tt$ into Equation (9), one obtains

$$W\left(\frac{r^2S}{4Tt}\right) = -0.5772 - \ln\frac{r^2S}{4Tt} = \ln\frac{2.25Tt}{r^2S} = 2.3\lg\frac{2.25Tt}{r^2S} \tag{12}$$

Additionally, a drawdown in an observation well that penetrates the main aquifer between elevations $z_1 = d'$ and $z_2 = d' + l'$ (Figure 1) can be obtained by averaging the drawdown in Equation (7) over this interval and can be written as [2,39,40]).

$$\begin{aligned}
s_{Ob} &= \frac{1}{l'}\int_{d'}^{l'+d'} s(r',z,t)dz \\
&= \frac{Q}{4\pi T}\left\{W(u) + \frac{2M}{\pi l}\sum_{n=1}^{\infty}\frac{1}{n}\left\{\sin\left[\frac{(l+d)n\pi}{M}\right] - \sin\left(\frac{n\pi d}{M}\right)\right\}\cdot\frac{1}{l'}\int_{d'}^{l+d'}\cos\left(\frac{n\pi z}{M}\right)dz\cdot W(u,x_n)\right\}
\end{aligned} \tag{13}$$

Substituting Equation (7) into Equation (11) results in

$$s_{Ob} = \frac{Q}{4\pi T}\left[W(u) + f_{r'}\left(u,\frac{r'}{M},\frac{l}{M},\frac{d}{M},\frac{l'}{M},\frac{d'}{M}\right)\right] \tag{14}$$

where

$$\begin{aligned}
f_{r'} &= \frac{2M^2}{\pi^2 ll'}\sum_{n=1}^{\infty}\frac{1}{n^2}\left\{\sin\left[\frac{(l+d)n\pi}{M}\right] - \sin\left(\frac{n\pi d}{M}\right)\right\} \\
&\cdot\left\{\sin\left[\frac{(l'+d')n\pi}{M}\right] - \sin\left(\frac{n\pi d'}{M}\right)\right\}\cdot W\left(u,\frac{n\pi r'}{M}\right)
\end{aligned} \tag{15}$$

in which $r'$ refers to the radial distance between the observation well and the pumping well; $l'$ is the well screen length of the observation well; $d'$ is the vertical distance from the bottom of the overlain imperious layer to the top of the well screen of the observation well; $Ss$ represents the specific storage of aquifer. It should be noted that $f_{r'}$ is a constant for a given $l'$, $l$, $r'$, $d$, $d'$, and $M$.

In addition, using Equations (9) and (10), the drawdown in an observation well of Equation (13) becomes

$$s_{Ob} = \frac{Q}{4\pi T}\left[2.3\lg\frac{2.25Tt}{r'^2S} + f_{r'}\left(\frac{r'}{M},\frac{l}{M},\frac{d}{M},\frac{l'}{M},\frac{d'}{M}\right)\right] \tag{16}$$

$$\begin{aligned}
f_{r'} &= \frac{4M^2}{\pi^2 ll'}\sum_{n=1}^{\infty}\frac{1}{n^2}\left\{\sin\left[\frac{(l+d)n\pi}{M}\right] - \sin\left(\frac{n\pi d}{M}\right)\right\} \\
&\cdot\left\{\sin\left[\frac{(l'+d')n\pi}{M}\right] - \sin\left(\frac{n\pi d'}{M}\right)\right\}\cdot K_0\left(\frac{n\pi r'}{M}\right)
\end{aligned} \tag{17}$$

### 2.2. Drawdown Solution for Flow in a Confined Aquifer within a Fully Penetrated Waterproof Curtain

A well located in a rectangular confined aquifer bounded by four man-made impermeable boundaries (the drop waterproof curtain) is illustrated on plan view in Figure 2. The drawdown solution for the fully bounded aquifer can be derived by the use of the method of images and superposition of solutions for an infinite confined aquifer. Owing to the four fully impervious boundaries, a set of $N$ mirror/fictitious wells can be treated as pumping wells. Thus, the final solution can be obtained as

$$s = \frac{Q}{4\pi T}\left(2.3\lg\frac{2.25Tt}{r_0^2S} + f_{r_0}\right) + \frac{Q}{4\pi T}\left(2.3\lg\frac{2.25Tt}{r_1^2S} + f_{r_1}\right) + \cdots + \frac{Q}{4\pi T}\left(2.3\lg\frac{2.25Tt}{r_n^2S} + f_{r_n}\right) \tag{18}$$

where $r_0$ is the radial distance from the observation well to the pumping well, $r_i$ ($i = 1, 2, \ldots, n$) is the distance of an observation well from the image well $i$.

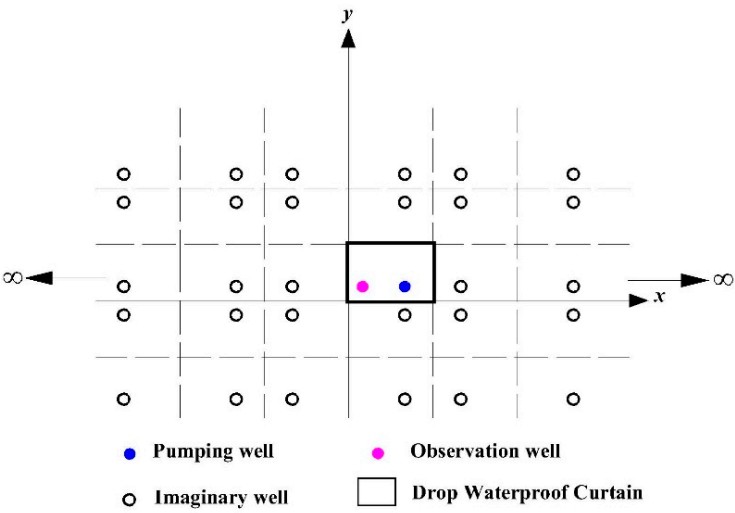

**Figure 2.** Plan view of a fully penetrated waterproof curtain.

For the purpose of illustration, Equation (18) can be rewritten as

$$s = \frac{Q}{4\pi T}\left(2.3\sum_{n=0}^{\infty}\lg\frac{2.25Tt}{r_n{}^2 S} + \sum_{n=0}^{\infty} f_{r_n}\right) = \frac{Q}{4\pi T}\left(2.3\lg\frac{(2.25Tt)^{n+1}}{r_0{}^2 r_1{}^2\cdots r_n{}^2 (S)^{n+1}} + \sum_{n=0}^{\infty} f_{r_n}\right)$$
$$= \frac{2.3Q}{4\pi T}\lg\frac{(2.25Tt)^{n+1}}{R^2(S)^{n+1}} + \frac{AQ}{4\pi T} \tag{19}$$

where $A = \sum_{n=0}^{\infty} f_{r_n}$ and one can easily see that $A$ is a constant; $R^2 = r_0{}^2 r_1{}^2 \cdots r_n{}^2$.

Most importantly, if the values of $Q$, $T$, and $Ss$ are known, Equation (19) can be further expressed as

$$s = \frac{2.3(n+1)Q}{4\pi T}\lg t + \frac{2.3Q}{4\pi T}\lg\frac{(2.25T)^{n+1}}{R^2(S)^{n+1}} + \frac{AQ}{4\pi T} \tag{20}$$

Obviously, Equation (18) is time-independent, and it can be found from Equation (18) that the drawdown ($s$) versus logarithmic time (lg t) curve is linear, as shown in Figure 3. The slope of the straight line ($i$) is given by

$$i = \frac{2.3(n+1)Q}{4\pi T} \tag{21}$$

The interception point with the time ($t$) axis has the coordinates $s = 0$ and $t = t_0$. Substitution these values into Equation (18), one can obtain

$$t_0 = \frac{SR^{\frac{2}{n+1}}}{2.25T10^{\frac{A}{2.3(n+1)}}} \tag{22}$$

Thus, the straight-line method may be helpful to determine the hydraulic parameter.

### 2.3. Estimation for Hydrogeologic Parameters

In this section, the steps of the procedure for the straight-line method are given as follows [2,28]:

Step 1: Plot the observed drawdown data on single logarithmic paper ($t$ on logarithmic scale) and construct the ultimate straight line and extend it to the zero-drawdown axis.

Step 2: Determine the geometric slope ($i$) of the ultimate straight line and also determine the interception point with the time axis where $s = 0$. Read the value of $t_0$.

Step 3: Using the known values of $Q$ and $i$, calculate the value of the $T$.

$$T = \frac{2.3(n+1)Q}{4\pi i} \tag{23}$$

Step 4: Calculate the value of constant $A$ in Equation (17). A few terms of the series involved are generally sufficient.

Step 5: Calculate the value of storage coefficient from Equation (20), and the calculation equation can be written as

$$S = \frac{2.25Tt_0 10^{\frac{A}{2.3(n+1)}}}{(R)^{\frac{2}{n+1}}} \tag{24}$$

Step 6: If necessary, one can use the following equations to calculate the values of hydraulic conductivity and specific storage, respectively.

$$K = \frac{T}{M} \tag{25}$$

and

$$S_s = \frac{S}{M} \tag{26}$$

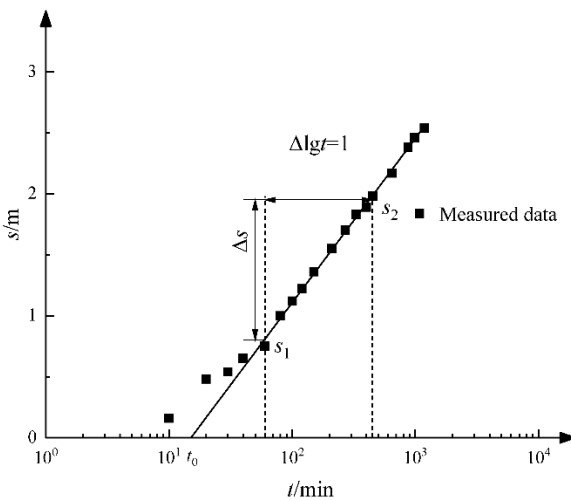

**Figure 3.** Schematic diagram of straight-line method.

## 3. Application for Parameter Estimation Using Field Test Data

### 3.1. Study Area

Pumping tests are often carried out to investigate the dewatering effect before the excavation in foundation pit engineering. The shape of the foundation pit is rectangular, and the drop waterproof curtain fully penetrates the main confined aquifer. The test site consists of (silty) fine sand with a thickness of 31.4m and is bounded by an impervious clay layer above and an impermeable bedrock. Two single pumping tests are performed in a foundation pit engineering in Wuhan, China, Figure 1 shows the plane position location of the pumping well (labeled W1) and observation wells (labeled G1 and G2). The pumping well (W1) and the observation wells H1–2 through G11–7 were installed within a confined aquifer to a depth of 18.4 m and 16.4m, and the screening interval of the three wells are 10 m long. Figure 4 shows the layout of pumping tests.

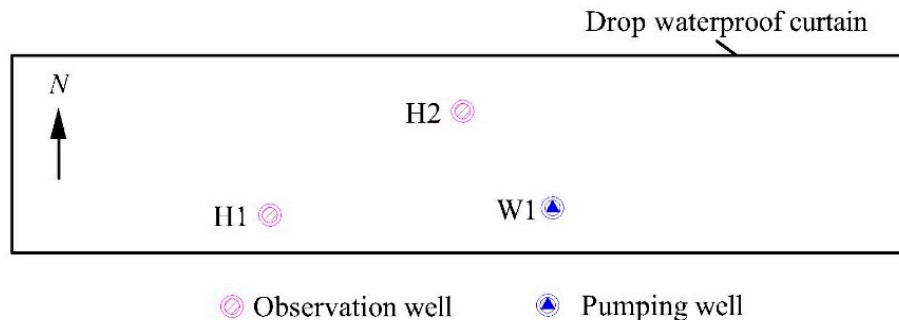

**Figure 4.** Schematic diagram straight-line method.

### 3.2. Field Pumping Tests

Two different single-well pumping tests are, respectively, performed at the field, in the two tests (No. 1 and No. 2), the pumping rates are maintained at 27.3 m$^3$/h and 28.0 m$^3$/h, respectively, and the radial distance $r_0$ from the observation well (H1) and H2 to pumping well (W1) are equal to 42 m and 26 m, respectively. The drawdown records for different single pumping tests are shown in Table 1.

**Table 1.** Observation drawdown data during pumping.

| Pumping Time (*t*/min) | Observation Well (H1) Drawdown (*s*/m) | Observation Well (H2) Drawdown (*s*/m) |
|---|---|---|
| 0 | 0 | 0 |
| 3 | 0.006 | 0.04 |
| 10 | 0.09 | 0.06 |
| 15 | 0.122 | 0.07 |
| 20 | 0.138 | 0.08 |
| 25 | 0.152 | 0.1 |
| 30 | 0.173 | 0.11 |
| 60 | 0.215 | 0.17 |
| 90 | 0.26 | 0.3 |
| 120 | 0.42 | 0.5985 |
| 150 | 0.58 | 0.7801 |
| 180 | 0.675 | 0.8504 |
| 210 | 0.84 | 1.0529 |
| 240 | 0.935 | 1.2037 |
| 270 | 0.997 | 1.2545 |
| 300 | 1.044 | 1.2681 |

### 3.3. Parameter Estimation

The known parameters are listed as: $M$ = 31.4 m, $l$ = 10 m, $d$ = 18.4 m, $d'$ = 16.4 m, $l/M$ = 0.32, when $d/M$ = 0.59 and $d'/M$ = 0.52, $f_r$ can be calculated by Equation (15), and the results are shown in Table 2.

**Table 2.** The values of $f_r$.

| $l/M$ | $r/M$ | | | | | | | | | |
|---|---|---|---|---|---|---|---|---|---|---|
| | **2** | **1** | **1/3** | **0.1** | **1/30** | **0.01** | **0.005** | **0.002** | **0.001** | **0.0005** |
| 0.1 | 0.00034 | 0.0130 | 0.4390 | 3.3949 | 8.6047 | 15.2123 | 19.1007 | 24.2574 | 28.1615 | 32.0661 |
| 0.3 | 0.0012 | 0.0383 | 0.5674 | 2.6123 | 5.6922 | 9.5087 | 11.7527 | 14.7297 | 16.9837 | 19.2382 |
| 0.5 | 0.0020 | 0.0630 | 0.8501 | 2.9352 | 5.3649 | 8.1824 | 9.8214 | 11.9919 | 13.6346 | 15.2775 |
| 0.7 | 0.0022 | 0.0702 | 0.9672 | 3.3991 | 6.1969 | 9.3944 | 11.2468 | 13.6980 | 15.5527 | 17.4075 |
| 0.9 | 0.0018 | 0.0570 | 0.7689 | 2.6914 | 4.9703 | 7.6265 | 9.1730 | 11.2214 | 12.7718 | 14.3223 |

Theoretically, there exists an infinity of reflection images, but in fact, a finite number of images (three or four) is enough for practical application [19], so three reflections are

chosen in this study. There are 20 imaginary wells for each single pumping test, and the distances between the observation well and imaginary wells are listed in Table 3.

**Table 3.** The distances of the image wells to the observation well.

| Group | $r_1$ | $r_2$ | $r_3$ | $r_4$ | $r_5$ | $r_6$ | $r_7$ | $r_8$ | $r_9$ | $r_{10}$ |
|---|---|---|---|---|---|---|---|---|---|---|
| No. 1 | 135.43 | 114.00 | 109.18 | 101.90 | 102.47 | 113.02 | 118.65 | 74.09 | 64.68 | 43.70 |
| No. 2 | 148.58 | 134.00 | 131.40 | 130.21 | 132.11 | 144.75 | 150.47 | 79.68 | 68.26 | 34.05 |
| | $r_{11}$ | $r_{12}$ | $r_{13}$ | $r_{14}$ | $r_{15}$ | $r_{16}$ | $r_{17}$ | $r_{18}$ | $r_{19}$ | $r_{20}$ |
| No. 1 | 57.71 | 66.38 | 98.75 | 267.42 | 257.23 | 255.13 | 252.11 | 252.34 | 256.80 | 259.33 |
| No. 2 | 31.16 | 40.78 | 76.04 | 241.96 | 233.30 | 231.81 | 231.14 | 232.22 | 239.63 | 243.13 |

The observed semilogarithmic plot for Observation Well H 1 is shown in Figure 5. It can be seen from Figure 5 that the slope of the straight-line ($i$) is 3.09 and $t_0$ = 61.24. The values of A and R in Equation (17) can be, respectively, calculated as 0.026 and $5.84 \times 10^{43}$. Following the above-mentioned procedure for the straight-line method, one can determine the parameters of the pumping confined aquifer, and the estimated hydraulic conductivity and storage coefficient are $K$ = 25.96 m/d and $S$ = 0.0011, respectively. In addition, in order to verify the correctness of the newly developed solution and the straight-line method, Figure 6 shows the drawdown predicted by the estimated hydraulic parameters and the observed values at relatively long pumping times, and the results are shown the reliability of the methods in this study.

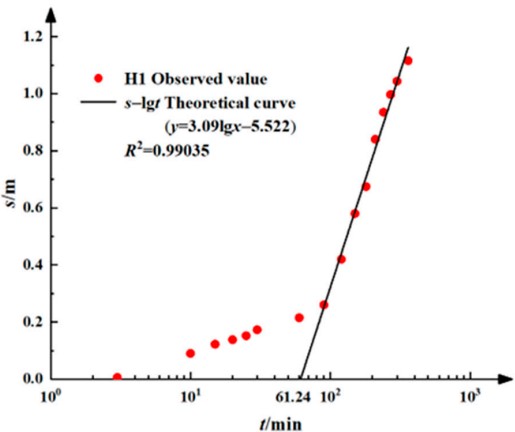

**Figure 5.** Drawdown versus semilogarithmic pumping time for observation well H1.

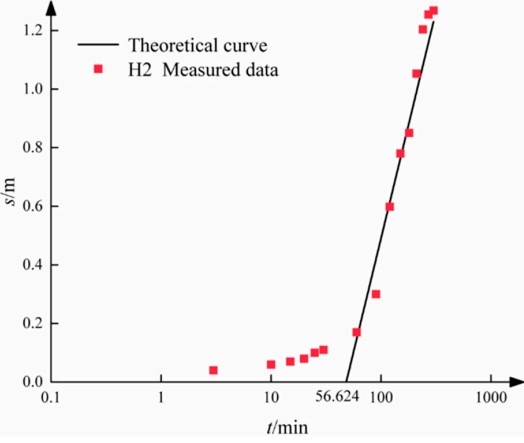

**Figure 6.** Comparison of the theoretical drawdown obtained by the newly estimated parameter and the measured field data for observation well H2.

## 4. Discussion

The newly developed solution is able to be applied to determining the hydraulic parameters using the collected drawdown data in field pumping tests. If necessary, the presented solution can be applied to analyze the drawdown responses in this fully bounded rectangular confined aquifer, to investigate the influence of the impervious boundary on the drawdown. What is more, it could show the application value in other areas, especially in excavation dewatering engineering.

However, some limitations of this study are needed to be addressed as follows. First of all, the obtained solution is derived from the solution of Hantush (1964) [2], thus, some effects in well hydraulics such as anisotropy, wellbore storage, skin effect, and leakage effect are not taken into consideration. Second, the presented straight-line method is only available when the pumping time is long enough, the method using the data during the whole pumping period needs to be discussed in the near future. What is more, the shape of the fully penetrated waterproof curtain is rectangular, which is commonly encountered in practice, but the other shaped waterproof curtain could be considered if necessary. Additionally, the pumping-induced flow is assumed to be Darcy, but the flow may be non-Darcy nearing the pumping well when the discharge is large, so the non-Darcy effect cannot be considered as well. Finally, the constant-head pumping test that is usually performed in the project of deep excavation dewatering may also not be addressed in this study. It is fortunate to investigate the above-mentioned points following the procedure in this study in the future.

## 5. Conclusions

In this study, the closed-form solutions for constant-rate pumping in a finite confined aquifer bounded by a rectangular-shaped fully penetrated waterproof curtain are obtained using the method of the image together with the principle of superposition. The pumping well is of partial penetration. A straight-line method of parameter estimation based on the new solution is proposed, and the case study for parameter estimation is used for the application of the solution and the method. The developed analytical solutions can be applied to estimate aquifer hydraulic parameters, evaluate boundary effects, and assist in the design of the foundation pit dewatering.

**Author Contributions:** Conceptualization, Y.L. and W.X.; methodology, B.P.; software, Y.L.; validation, H.W. and B.P.; formal analysis, Y.L.; investigation, W.X.; resources, F.X.; data curation, C.Z.; writing—original draft preparation, Y.L.; writing—review and editing, F.X.; visualization, Y.L.; supervision, F.X.; project administration, Y.L.; funding acquisition, F.X. All authors have read and agreed to the published version of the manuscript.

**Funding:** This work is financially supported by the Scientific research project of Powerchina Hubei Electric Engineering Co., Ltd. (Grant No. K2021-2-04) and the national natural science foundation of China (Grant No. 52209148).

**Data Availability Statement:** All data included in this study are available upon request by contact with the corresponding author.

**Conflicts of Interest:** The authors declare no conflict of interest.

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
