# Peer review of "Hydrogeological Parameter Estimation of Confined Aquifer within a Rectangular Shaped Drop Waterproof Curtain"

_water, doi:10.3390/w15020356_

Round 1

Reviewer 1 Report

This manuscript presents a analytical solution for a constant discharge pumping in a confined aquifer within a rectangular-shaped drop waterproof curtain.  This new solution is an extension of Hantushs (1964) solution by making use of the image method coupled with superposition principle for the rectangular-shaped drop waterproof curtain (aquifer with impermeable boundaries in x and y directions).  I suggest minor revision of this manuscript which needs some corrections and more explanations as follows:

1. Check if the cited articles are all included in the reference, i.e., Doherty (2018)

2. Explain if Eqs. (16)-(17) were derived by the authors or presented before in other article?

3. The function, Ko(), needs to be explained in the text.

4. Check the correctness of equation number indicated in the text 

Author Response

This manuscript presents an analytical solution for a constant discharge pumping in a confined aquifer within a rectangular-shaped drop waterproof curtain.  This new solution is an extension of Hantush‘s (1964) solution by making use of the image method coupled with superposition principle for the rectangular-shaped drop waterproof curtain (aquifer with impermeable boundaries in x and y directions).  I suggest minor revision of this manuscript which needs some corrections and more explanations as follows:

  1. Check if the cited articles are all included in the reference, i.e., Doherty (2018)

Reply: The cited study of Doherty (2018) has been added in the ‘References’ part.

  1. Explain if Eqs. (16)-(17) were derived by the authors or presented before in other article?

Reply: Eqs. (16)-(17) were derived on the basis of Eqs. (9), (10) and (13), we have added the study of Hantush (1964) herein.

  1. The function, Ko(), needs to be explained in the text.

Reply: The explanation of Ko() has been added in the main text.

  1. Check the correctness of equation number indicated in the text

Reply: We have carefully checked the correctness of equation number,  and the misuse has been corrected.

Reviewer 2 Report

It seems that the paper has a good structure. It has a good theoretical basis. Acceptable results are presented and they are validated by field data. In my opinion, the paper can be accepted after minor revision.

Keywords: Include the name of a country or multi-country region, when appropriate.

Line 57-59: Add the following reference:

https://doi.org/10.1007/s10064-022-02612-3

Lines 59-81: Add the following reference:

https://doi.org/10.1007/s10668-022-02368-6

Line 119: If the pumping time is long or r is too small, the value of u is less than 0.01 and then equation 10 can be used.

Line 134: The definition of r given here is different from the r discussed in the well function equation. It is better to separate them.

The discussion gives an interpretation of the results and their significance and limitations, with reference to work by other authors.

Pay attention to the journal's instructions for preparing the reference list. Different formats are seen.

Author Response

It seems that the paper has a good structure. It has a good theoretical basis. Acceptable results are presented and they are validated by field data. In my opinion, the paper can be accepted after minor revision.

Keywords: Include the name of a country or multi-country region, when appropriate.

Reply: We have clarified the field data collected from Wuhan, China in ‘Abstract’.

Line 57-59: Add the following reference:

https://doi.org/10.1007/s10064-022-02612-3

Reply: Added.  See reference [1].

Lines 59-81: Add the following reference:

https://doi.org/10.1007/s10668-022-02368-6

Reply: Added.  See reference [2].

Line 119: If the pumping time is long or r is too small, the value of u is less than 0.01 and then equation 10 can be used.

Reply: Thanks.  We have revised it as suggested.

Line 134: The definition of r given here is different from the r discussed in the well function equation. It is better to separate them.

Reply: Thanks.  We have revised it as suggested.

Pay attention to the journal's instructions for preparing the reference list. Different formats are seen.

Reply: Revised.